# Improved Overparametrization Bounds for Global Convergence of SGD for Shallow Neural Networks

**Bartłomiej Polaczyk** *b.polaczyk@mimuw.edu.pl*
*Institute of Informatics, University of Warsaw*

**Jacek Cyranka** *cyranka@mimuw.edu.pl*
*Institute of Informatics, University of Warsaw*

**Reviewed on OpenReview:** *https://openreview.net/forum?id=RjZq6W6FoE*

## Abstract

We study the overparametrization bounds required for the global convergence of stochastic gradient descent algorithm for a class of one hidden layer feed-forward neural networks equipped with ReLU activation function. We improve the existing state-of-the-art results in terms of the required hidden layer width. We introduce a new proof technique combining nonlinear analysis with properties of random initializations of the network.

## 1 Introduction

The study of convergence properties of mini-batch stochastic gradient descent (SGD) iterations applied to feed-forward neural nets (NN) is at the core of modern machine learning research. SGD with its variants like ADAM is the most common optimization scheme applied for supervised training of NN. In principle however, the loss landscape encountered when training NN is highly nonconvex, especially for deep nonlinear NN as revealed, e.g., by visualizations performed by Li et al. (2018), and construction proofs of spurious local minima by Auer et al. (1996a); Brutzkus et al. (2018). The nonconvexity may have severe consequences for practical NN training routines, as SGD may potentially get stuck at a spurious local minimum or a saddle point and cease to converge further down the loss valley. Yet, practice suggests that with enough overparametrization, SGD iterations achieve global minima (i.e., perfect fitting of the training data) most of the times. This phenomenon is not fully understood yet and is the main theme of this paper.

Contemporary research on NN convergence theory was initiated with the study of linear networks. The loss landscape in this setting was fully characterized by Kawaguchi (2016), solving the problem stated by Choromanska et al. (2015). The research revealed the feasibility of global SGD convergence for deep NN despite the loss landscape nonconvexity.

Even though it seems difficult to fully characterize the loss landscape in the nonlinear setting, proving the global SGD convergence is still feasible. Recent research suggests that SGD converges globally with high probability for random initialization of weights, under the assumption of sufficiently large overparametrization expressed in terms of NN layers' widths. The first result of this kind required an unrealistic level of overparametrization of polynomial order in the number of samples, cf. Allen-Zhu et al. (2019). The following series of related results (see Table 1) further reduced the required level of overparametrization using various techniques and assumptions on training data. Especially in the case of Deep NN equipped with analytic activation functions, an overparametrization of the linear order with respect to the number of training examples is sufficient. However, such tight overparametrization results do not apply in the case of a non-differentiable ReLU activation function (see Table 1). Existing theoretical bounds still require a significantly larger number of parameters than used in practice. The question about an exact boundary marking the minimal number of parameters required for the global convergence is still open even for shallow (one hidden layer) ReLU NN, see, e.g., Oymak & Soltanolkotabi (2020).

## 1.1 Main Contribution.

We establish a new theoretical order of overparametrization required for SGD convergence towards a global minimizer for one hidden layer NN with ReLU activations, improving known state-of-the-art bounds. We introduce a new proof technique based on nonlinear analysis. First, we show the global convergence of continuous solutions of the differential inclusion (DI) being a nonsmooth analog of the gradient flow for the MSE loss. Second, using the existing nonsmooth analysis results, we establish closeness of continuous trajectories to SGD sequences until convergence for a sufficiently small learning rate.

The concept of studying the dynamics of continuous solutions pursued in this work already appeared earlier (Arora et al., 2019a; Du et al., 2019b). However, the authors treated the convergence of SGD sequences independently from the analysis of continuous solutions, which served motivational purpose only. We develop a rigorous method for establishing the convergence of SGD sequences via the convergence of continuous solutions, which works for general nonsmooth approximators including deep NN and general loss functions.

## 1.2 Informal statements.

We derive our global convergence results under the following assumptions and notation (made precise later on). We consider a shallow (one hidden layer) NN equipped with ReLU activation in the hidden layer and linear activation in the output layer. Let $N$ be the sample size and denote $d_0, d_1, d_2$ the dimensions of the input, hidden layer, and output, respectively. The input data comes from the i.i.d. sub-Gaussian distribution on the sphere in $\mathbb{R}^{d_0}$, where $d_0 \in [N^{\delta_0}, N]$ for some $\delta_0 \in (0, 1)$. The initial weight vector $\theta_0$ (being a vector obtained by flattening the weights of all NN layers) is obtained via the LeCun scheme (variance scales with width). $\mathcal{L}(\theta)$ is the MSE loss for some label matrix and a weight vector $\theta$. The subdifferential in the sense of Clarke is denoted by $\partial$ and $\tilde{\Omega}$ is the $\Omega$ notation hiding the logarithmic terms. All presented results hold with high probability (WHP), meaning that the probability of the event converges to one as the number of samples $N$ diverges to infinity, a convention widely adopted in the literature.

Our first main result provides a condition for the global convergence of the continuous solutions of the nonsmooth analog of gradient flow for $\mathcal{L}$.

**Theorem 1.1** (Informal Corollary 4.4). *Let the width of the shallow NN satisfy $d_1 = \tilde{\Omega}(N^{1.25})$. Then, any solution $\theta \colon \mathbb{R}_+ \to \mathbb{R}$ to the DI Cauchy problem $\theta(0) = \theta_0$, $\dot{\theta}(t) \in -\partial\mathcal{L}(\theta(t))$ satisfies $\mathcal{L}(\theta(t)) \leq \mathcal{L}(\theta(0)) \exp(-ctd_1)$ for all $t \geq 0$ and some constant $c > 0$ WHP.*

The second main result establishes the global convergence for the mini-batch SGD iterates WHP.

**Theorem 1.2** (Informal Theorem 5.1). *Let the width of the shallow NN satisfy $d_1 = \tilde{\Omega}(N^{1.25})$. Then, for any error $\varepsilon > 0$ and any mini-batch size, the mini-batch SGD sequences with step size small enough achieve the loss value below $\varepsilon$ at a linear convergence rate WHP.*

We obtain Theorem 1.2 via the following result. It is stated for general approximators (including deep ReLU NN) and general losses (including hinge loss, cross-entropy etc.). We believe it is of independent interest. We drop the assumption on the MSE loss and particular NN, and use the notion of an arbitrary loss $\tilde{\mathcal{L}}$.

**Theorem 1.3.** *(Informal Theorem 5.6) Let the loss function $\tilde{\mathcal{L}}$ be arbitrary satisfying some mild technical conditions. Additionally, assume there exists a nonempty compact set $Q$, s.t. any solution $\theta$ to the DI $\dot{\theta}(t) \in -\partial\tilde{\mathcal{L}}(\theta(t))$ if initialized in $Q$, remains in some compact set $G$ and converges to zero as $\tilde{\mathcal{L}}(\theta(t)) \leq \tilde{\mathcal{L}}(\theta(0))e^{-\gamma t}$. Then, for any $\varepsilon > 0$, the SGD sequences initialized in $Q$ and with step size small enough achieve the loss value below $\varepsilon$ at a linear convergence rate WHP.*

Let us comment briefly on some key aspects of our results.

**Overparametrization Bound Improvement.**

Theorem 1.2 improves state-of-the-art overparametrization bounds for global SGD convergence for shallow ReLU NN – in Table 1 we compare it to the selected works that we find most related. For instance, Nguyen (2021) require $d_1 = \Omega(N^2)$. Similarly, Oymak & Soltanolkotabi (2020) require $d_1 = \Omega(N^4/d_0^3)$

(which is better for $d_0$ larger than $N^{2/3}$), where they train the first weight matrix only and the second weights matrix remains fixed, cf. Remarks 5.3 and 5.4 for a detailed discussion and Section 6 for numerical experiments comparing both setups. We also note that we have more general data assumptions than Oymak & Soltanolkotabi (2020).

On the other hand, results from Kawaguchi & Huang (2019) and Liu et al. (2022) require only linear overparametrization and work for more general data. However, they do not apply to ReLU as they rely heavily on the smoothness of the activation function. In particular, analysis of non-smooth activation functions seems to be a much more challenging task, see e.g., a result showing the existence of spurious local minima in the ReLU setting Safran & Shamir (2018).

**Discrete vs Continuous Convergence.**

The idea of establishing a link between continuous solutions to the gradient flow and their discrete GD analogs for deep linear networks was introduced recently by Elkabetz & Cohen (2021). Their method require the Hessian to exist and to be bounded along the continuous trajectories. Such approach does not work when a nonsmooth activation function, e.g. ReLU, is employed – we provide additional evidence supporting this claim in Section 6. Our approach of passing from continuous solutions to SGD sequences is more general because it works in the differential inclusions setting, which treats nondifferentiable objectives (in contrast to Elkabetz & Cohen (2021)).

**SGD step size.**

One should keep in mind that Theorem 1.2 is qualitative – it does not provide a constructive condition for the step size to guarantee convergence. However, existing quantitative results for ReLU NNs provide global convergence of the SGD scheme under the condition that the learning rate is of order $\mathcal{O}(1/N^2)$, which is still far from the learning rates used in ML practice. Cf., e.g., Arora et al. (2019b); Oymak & Soltanolkotabi (2020); Nguyen (2021).

Table 1: A Perspective on related work. Reported results using notation $\widetilde{\Omega}$ hides logarithmic terms, $N$ is the number of train samples, $d_0$ is the input dimension, $L$ is the number of layers of deep NN

| Work | Algorithm | ReLU | Deep | Data | Scaling | Initialization |
|------|-----------|------|------|------|---------|----------------|
| Du et al. (2019a) | GD | no | yes | non degenerate normalized | $\tilde{\Omega}(2^{O(L)}N^4)$ | LeCun |
| Kawaguchi & Huang (2019) | GD | no | yes | normalized | $\tilde{\Omega}(Nd_0)$ (shallow) $\tilde{\Omega}(N + d_0L^2)$ (deep) | LeCun |
| Daniely (2020b) | GD | no | no | gaussian | $\tilde{\Omega}(N/d_0)$ | LeCun |
| Liu et al. (2022) | SGD | no | yes | non degenerate normalized | $\tilde{\Omega}(N)$ | LeCun |
| Allen-Zhu et al. (2019) | SGD | yes | no | separable | $\tilde{\Omega}(N^{24}L^{12})$ | LeCun |
| Arora et al. (2019b) | GD | yes | yes | unif. on sphere | $\tilde{\Omega}(N^7)$ | LeCun |
| Zou & Gu (2019) | SGD | yes | yes | separable | $\tilde{\Omega}(N^8L^{12})$ | LeCun |
| Daniely (2020a) | SGD | yes | no | unif. on sphere | $\tilde{\Omega}(N/d_0)$ | non-standard[1] |
| Oymak & Soltanolkotabi (2020) | SGD (on layer 1) | yes | no | unif. on sphere | $\tilde{\Omega}(N^4/d_0^3)$ | LeCun |
| Nguyen (2021) | GD | yes | yes | subgaussian | $\tilde{\Omega}(N^2)$ (shallow) $\tilde{\Omega}(N^3)$ (deep) | LeCun |
| **Ours** | SGD | yes | no | subgaussian | $\tilde{\Omega}(N^{1.25})$ | LeCun |

## 1.3 Other Related Work.

We summarize the current literature concerning the question of SGD global convergence for NN equipped with the MSE loss in Table 1. We split the results into two groups, first the ones working for smooth acti-

---

[1]Half of the weights of the first layer is initialized as standard Gaussians and the second half is a copy of the first half. Half of the weights of the second layer equals to some big enough fixed real number, while the other half equals to the negative of that number.

vations and second, the results for ReLU activation function, also considered in this work. Similar and, in some cases, tighter overparametrization results have been established for training deep NN equipped with cross-entropy loss Li & Liang (2018); Ji & Telgarsky (2020); Chen et al. (2021). All existing results except Daniely (2020b) and Daniely (2020a) from Table 1 are derived under the assumption that there is a significant overparametrization of the NN under study (at least one wide hidden layer). Daniely (2020a) provides an optimal bound but requires weights initialization that is far from standard, making the optimization process equivalent to NTK optimization. Daniely (2020b) provides near-optimal bound (with an extra logarithmic factor), which however does not work for ReLU. Earlier work focused on the non-existence of spurious local minima without consideration of SGD dynamics Xie et al. (2017). The extreme case of overparametrization, i.e., infinite layer width, has also been analyzed in Chizat & Bach (2018); Jacot et al. (2018); Mei et al. (2018).

One can also find negative results in the literature, demonstrating, e.g., the existence of spurious local minima in underparameterized regimes, Auer et al. (1996b), or convergence towards spurious local minima, as in Brutzkus et al. (2018). As for other fundamental properties, nonlinear NN are universal approximators Cybenko (1989); Shaham et al. (2018). NN memorization property has also been extensively studied – in the case of shallow NN, known overparametrization bounds for perfect memorization of the data are near-optimal Zhang et al. (2017); Hardt & Ma (2017); Nguyen & Hein (2018); Baldi & Vershynin (2019); Yun et al. (2019); Bubeck et al. (2020).

### 1.4 Organization of this paper

In Section 2 we introduce the notation and recall some facts regarding differential inclusions. In Section 3 we study the properties of the DI solutions for MSE loss. In Section 4 we prove the global convergence result for DI solutions under random initialization. In Section 5 we extend the result of Section 4 to SGD iterates. In Section 6 we present some numerical experiments related to our results. We summarize our findings in Section 7.

## 2 Preliminaries

Let $X \in \mathbb{R}^{N \times d_0}$ be a matrix of the training inputs (arranged rowwise) and $Y \in \mathbb{R}^{N \times d_2}$ be a matrix of training labels, where $N \in \mathbb{N}_+ \overset{\text{def}}{=} 1, 2 \ldots$ is the sample size and $d_0, d_2 \in \mathbb{N}_+$ are the dimensions of the input and output respectively. Consider the following one hidden-layer feed-forward NN

$$\hat{Y} \overset{\text{def}}{=} \phi(XW)V,$$

where for some $d_1 \in \mathbb{N}_+$, $W \in \mathbb{R}^{d_0 \times d_1}$ and $V \in \mathbb{R}^{d_1 \times d_2}$ are the weight matrices and $\phi \colon \mathbb{R} \to \mathbb{R}$ is the ReLU activation function applied element-wise. We often assume that $X, Y$ are fixed and known from context, whence they are not explicitly mentioned as parameters, e.g., in the loss function formula. We denote the hidden layer by $H$, i.e., $H \overset{\text{def}}{=} \phi(XW) \in \mathbb{R}^{N \times d_1}$. We write $D \overset{\text{def}}{=} d_0 d_1 + d_1 d_2$ and denote parameter vector by $\theta \in \mathbb{R}^D$, i.e., $\theta$ is obtained by stacking vectorized matrices $W, V$. We identify matrices with their vectorized forms and write simply $\theta = (W, V)$.

The standard dot product and Euclidean distance on $\mathbb{R}^d$ for $d \in \mathbb{N}_+$ are denoted by $\langle \cdot, \cdot \rangle$ and $\| \cdot \|$. For $x \in \mathbb{R}^d$ and $r > 0$, $B(x, r) \overset{\text{def}}{=} \{ y \in \mathbb{R}_d \colon \|y - x\| \le r \}$ is the closed ball with radius $r$ centered at $x$. For a matrix $A \in \mathbb{R}^{n_r \times n_c}$, $A_{i:}$ denotes the $i$-th row vector of $A$ for $i \in [n_r]$, and $A_{:i}$ denotes the $i$-th column vector of $A$ for $i \in [n_c]$, where $[k] \overset{\text{def}}{=} \{1, \ldots, k\}$ for $k \in \mathbb{N}_+$. Finally, we denote the minimal eigen- and singular values of $A$ by $\lambda_{min}(A)$ and $\sigma_{min}(A)$, i.e., $\sigma_{min}(A) = \sqrt{\lambda_{min}(A^T A)}$, while the operator and Frobenius norms of $A$ are denoted by $\|A\|_{op}$ and $\|A\|_F$.

Our aim is to optimize the MSE loss function $\mathcal{L} \colon \mathbb{R}^D \to \mathbb{R}_+$, defined via $\mathcal{L}(\theta) \overset{\text{def}}{=} \frac{1}{2} \|Y - \hat{Y}\|_F^2$. The widely applied ReLU activation function is non-differentiable at $x = 0$ but the generalized derivative in the sense of Clarke, cf. Clarke (1983), exists and is equal to the interval $[0, 1]$. More precisely, let $d \in \mathbb{N}_+$ and $f \colon \mathbb{R}^d \to \mathbb{R}$ be a locally Lipschitz function. By Rademacher's theorem, the set $D_f$ of points where $f$ is differentiable is of full Lebesgue measure. For $x \in D_f$, we denote the gradient of $f(x)$ by $\nabla f(x)$. Then, the Clarke

subdifferential of $f$ at any point $x \in \mathbb{R}^d$ is defined via

$$\partial f(x) \stackrel{\text{def}}{=} \text{conv} \left\{ y \in \mathbb{R}^d : \quad y = \lim_{n \to \infty} \nabla f(x_n) \quad \text{for some} \quad D_f \ni x_n \stackrel{n \to \infty}{\to} x \right\}$$

where conv is the convex hull operator. The Clarke subdifferential satisfies $\partial f(x) = \{\nabla f(x)\}$ for any $x \in D_f$. We refer the reader to Rockafellar & Wets (2009) for a detailed treatment of generalized gradients.

Recall that a curve[2] $x \colon \mathbb{R}_+ \to \mathbb{R}^d$ is absolutely continuous if there exists a map $v \colon \mathbb{R}_+ \to \mathbb{R}^d$ that is integrable on compact intervals and s.t. $x(t) - x(0) = \int_0^t v(s)\, ds$ for all $t \geq 0$. To lighten the notation we sometimes write $\frac{d}{dt} x(t) = \dot{x}(t)$ and call any absolutely continuous curve an arc. For a given $x_0 \in \mathbb{R}^d$ and $f \colon \mathbb{R}^d \to \mathbb{R}$, we are interested in finding arcs $x$ that are solutions to the following differential inclusion Cauchy problem

$$x(0) = x_0, \quad \dot{x}(t) \in -\partial f(x(t)) \quad \text{for a.e. } t \geq 0. \tag{1}$$

The following property plays a crucial role in analyzing such problems – we say that $f$ satisfies the *chain rule* if for any arc $x \colon \mathbb{R}_+ \to \mathbb{R}^d$,

$$\left\{ \langle v, \dot{x}(t) \rangle : v \in \partial f(x(t)) \right\} = \left\{ \frac{d}{dt}(f \circ x)(t) \right\} \quad \text{for a.e. } t \geq 0. \tag{2}$$

Consider the dynamics given by the following DI obtained from equation 1 by taking $f = \mathcal{L}$,

$$\theta(0) = \theta_0, \quad \dot{\theta}(t) \in -\partial \mathcal{L}(\theta(t)) \quad \text{for a.e. } t \geq 0, \tag{3}$$

where $\theta_0 \in \mathbb{R}^D$ is some initial value. Note that a-priori it is unknown if there exists a solution to equation 3 defined on the whole interval $[0, \infty)$. Recall the notation $H = \phi(XW)$. The following standard result is due to the fact that $\mathcal{L}$ satisfies the chain rule equation 2, cf. Davis et al. (2020), combined with usual arguments regarding DIs, subdifferential of $\mathcal{L}$ and Grönwall's lemma. Since we were unable to find such statement that rigorously treats its existential component connected to the theory of DIs, we provide its detailed proof in Appendix A.

**Proposition 2.1.** *For any initial $\theta_0 \in \mathbb{R}^D$, there exists $T > 0$ and a solution $\theta \colon [0, T) \to \mathbb{R}^D$ to the DI equation 3. Moreover, for any bounded domain $G \ni \theta_0$, each solution $\theta$ to equation 3 can be extended to infinity or up until it hits the boundary of $G$. Finally, for any such $\theta$, denoting $\alpha_0(s) \stackrel{\text{def}}{=} \sigma_{min}(H^T(\theta(s)))$, one gets*

$$\mathcal{L}(\theta(t)) \leq \mathcal{L}(\theta(0)) \exp\left( -2 \int_0^t \alpha_0^2(s)\, ds \right) \quad \text{for every } t \in [0, T).$$

## 3 Dynamics of the Differential Inclusion

In this section, we show that the integral of the loss (square root) along the parameter $\theta$ trajectories determined by the DI equation 3 satisfies a simple one-dimensional differential inequality. From that we infer boundedness properties of the loss along trajectories. The constants appearing in the differential inequality depend on the initialization properties only which allows us to provide WHP estimates in Section 4.

Recall the notation $H = \phi(XW)$ and $\alpha_0(s) = \sigma_{min}(H^T(\theta(s)))$. By Weyl's inequality, cf., e.g., (Dax, 2013, Theorem 4), and Lemma H.1,

$$|\alpha_0(t) - \alpha_0(0)| \leq \|H(t) - H(0)\|_F \leq \|X(W(t) - W(0))\|_F \leq \|X\|_{op}\|W(t) - W(0)\|_F. \tag{4}$$

Therefore, to use Proposition 2.1, in lemma below we bound the quantity $\|X\|_{op}\|W(t) - W(0)\|_F$. We defer its proof, which is based on a careful application of Grönwall's lemma, to Appendix B.

**Lemma 3.1.** *Any solution $\theta \colon [0, T) \to \mathbb{R}$, $\theta = (W, V)$, to the DI equation 3 satisfies*

$$\|\theta(t) - \theta(0)\| \leq \sqrt{2}\|X\|_{op}\big(\|W(0)\|_F + \|V(0)\|_F\big)\bar{\mathcal{L}}(t) \exp\big(\sqrt{2}\|X\|_{op}\bar{\mathcal{L}}(t)\big) \tag{5}$$

---

[2]We use the same symbols to denote points and curves.

*for every* $t \in [0, T)$, *where* $\bar{\mathcal{L}}(t) = \int_0^t \sqrt{\mathcal{L}(\theta(s))}\, ds$. *Moreover*

$$\|X\|_{op}\|W(t) - W(0)\|_F \leq \frac{1}{2}\Big(c_1\bar{\mathcal{L}}(t) + c_2\big(\bar{\mathcal{L}}(t)\big)^2\Big)\exp\Big(c\big(\bar{\mathcal{L}}(t)\big)^2\Big) \tag{6}$$

*for every* $t \in [0, T)$, *where*

$$c_1 \overset{\text{def}}{=} 2\sqrt{2}\|X\|_{op}^2\|V(0)\|_F, \quad c_2 \overset{\text{def}}{=} 2\|X\|_{op}^3\|W(0)\|_F, \quad c \overset{\text{def}}{=} \|X\|_{op}^2. \tag{7}$$

Using Lemma 3.1 and Proposition 2.1 we infer in the proposition below that loss trajectories along solutions to the DI equation 3 obey some specific differential inequality. This observation is crucial for obtaining the main results of this paper, i.e., Corollary 4.4 and Theorem 5.1.

**Proposition 3.2.** *Let* $c, c_1, c_2$ *be as in Lemma 3.1, equation 7. Set*

$$a \overset{\text{def}}{=} \sqrt{\mathcal{L}(\theta(0))}, \quad \alpha \overset{\text{def}}{=} \sigma_{min}(H^T(\theta(0))). \tag{8}$$

*If for some* $T > 0$, $\theta\colon [0, T) \to \mathbb{R}^D$ *solves the DI equation 3, then* $\bar{\mathcal{L}}(t) \overset{\text{def}}{=} \int_0^t \sqrt{\mathcal{L}(\theta(s))}\, ds$ *is a solution* $y\colon [0, T) \to \mathbb{R}$ *to the problem*

$$y(0) = 0; \quad y'(t) \leq a\exp\big(\alpha t(c_1 y(t) + c_2 y^2(t))e^{cy^2(t)} - \alpha^2 t\big) \ \text{for all} \ t \in [0, T). \tag{9}$$

*Proof.* Using Proposition 2.1, the inequality $(u - v)^2 \geq u^2 - 2u|v|$ for $u \geq 0$, $v \in \mathbb{R}$, and the estimate from equation 4, we get for all $t \in [0, T)$,

$$\sqrt{\mathcal{L}(\theta(t))} \leq \sqrt{\mathcal{L}(\theta(0))} \cdot \exp\Big(-\int_0^t \alpha_0^2(s)\, ds\Big)$$

$$\leq \sqrt{\mathcal{L}(\theta(0))} \cdot \exp\Big(-t\alpha_0^2(0) + 2\alpha_0(0)\int_0^t |\alpha_0(s) - \alpha_0(0)|\, ds\Big)$$

$$\leq \sqrt{\mathcal{L}(\theta(0))} \cdot \exp\Big(-t\alpha_0^2(0) + 2\alpha_0(0)\int_0^t \|X\|_{op}\|W(s) - W(0)\|_F\, ds\Big).$$

Using the bound from equation 6 due to Lemma 3.1 and noting that $\bar{\mathcal{L}}'(t) = \sqrt{\mathcal{L}(\theta(t))}$, we arrive at

$$\bar{\mathcal{L}}'(t) \leq a \cdot \exp\Big(-t\alpha^2 + 2\alpha\int_0^t \|X\|_{op}\|W(s) - W(0)\|_F\, ds\Big)$$

$$\leq a \cdot \exp\Big(-t\alpha^2 + \alpha\int_0^t \big(c_1\bar{\mathcal{L}}(s) + c_2\big(\bar{\mathcal{L}}(s)\big)^2\big)\exp\big(c\big(\bar{\mathcal{L}}(s)\big)^2\big)\, ds\Big)$$

and the conclusion follows by estimating $\bar{\mathcal{L}}(s) \leq \bar{\mathcal{L}}(t)$ for all $s \in [0, t]$. $\qquad\square$

Perhaps surprisingly, due to the double exponential dependence on $y^2(t)$, a simple condition involving $a, c, c_1, c_2, \alpha$ determines that solutions to equation 9 remain bounded by $2a/\alpha^2$ for all times, as demonstrated in Lemma 3.3 below. This property is illustrated in Figure 1.

**Lemma 3.3.** *Let* $a, \alpha, c, c_1, c_2$ *be some arbitrary parameters of equation 9. If* $\alpha > 0$ *and*

$$\Big(\frac{ac_1}{\alpha^3} + \frac{a^2 c_2}{\alpha^5}\Big)\exp\big(4ca^2/\alpha^4\big) < \frac{1}{8}, \tag{10}$$

*then for any* $T > 0$, *any solution* $y\colon [0, T) \to \mathbb{R}$ *to the problem from equation 9 is bounded from above by* $2a/\alpha^2$ *and its derivative at any time* $t \in [0, T)$ *is bounded by* $ae^{-\alpha^2 t/2}$.

*Proof.* Let $y\colon [0, T) \to \mathbb{R}$ be any solution to equation 9. Set

$$t_0 = \inf\big\{ t \in [0, T) \colon \ \big(c_1 y(t) + c_2 y^2(t)\big)e^{cy^2(t)} = \alpha/2 \big\}.$$

By assumption $y(0) = 0$ and $\alpha > 0$, whence by continuity of $y$, $t_0 > 0$. Moreover, for a.e. $t < t_0$,

$$y'(t) \le a \exp\left(\alpha t(c_1 y(t) + c_2 y^2(t))e^{cy^2(t)} - \alpha^2 t\right) \le a \exp\left(-\alpha^2 t/2\right),$$

whence $y(t) \le 2a/\alpha^2 \cdot (1 - e^{-\alpha^2 t/2}) < 2a/\alpha^2$ for all $t < t_0$. If $t_0 < T$, then by continuity of $y$ on $[0, T)$, $y(t_0) \le 2a/\alpha^2$ as well, in which case

$$\frac{1}{8} = \frac{1}{4\alpha}\left(c_1 y(t_0) + c_2 y^2(t_0)\right)e^{cy^2(t_0)} \le \left(\frac{ac_1}{\alpha^3} + \frac{a^2 c_2}{\alpha^5}\right)\exp\left(4ca^2/\alpha^4\right)$$

but this yields a contradiction with equation 10. Therefore $t_0 = T$ as desired. □

Using Proposition 3.2 in conjunction with Lemma 3.3, we obtain in the theorem below the announced global convergence guarantee for continuous parameter trajectories.

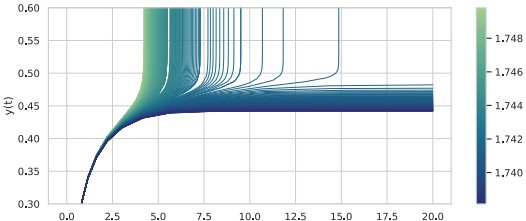

Figure 1: Numerical illustration of the solutions to equation 9 for fixed $a, c_1, c_2, c = 1$ and $\alpha \in [2.042, 2.045]$. The color scale encodes the values of $4(ac_1/\alpha^3 + 2a^2 c_2/\alpha^5)\exp(4ca^2/\alpha^4)$, the quantity determining equation 10. It is visible that the solution $y(t)$ either remains bounded or explodes rapidly depending on the condition involving the constants.

**Theorem 3.4.** *Let $a, \alpha, c, c_1, c_2$ be as in Proposition 3.2. Assume that $\alpha > 0$ and that at initialization*

$$F(\theta(0), X, Y) \stackrel{\text{def}}{=} \left(\frac{ac_1}{\alpha^3} + \frac{a^2 c_2}{\alpha^5}\right)\exp\left(\frac{4ca^2}{\alpha^4}\right) < \frac{1}{8}. \tag{11}$$

*Then, any solution $\theta \colon [0, T) \to \mathbb{R}^D$ to the DI equation 3 can be extended to a solution on $\mathbb{R}_+$ and any such extension satisfies for all $t \ge 0$,*

$$\mathcal{L}(\theta(t)) \le \mathcal{L}(\theta(0)) \exp(-t\alpha_0^2(0)) \tag{12}$$

*and for $u \stackrel{\text{def}}{=} 4\|X\|_{op}\sqrt{\mathcal{L}(\theta(0))}/\alpha_0(0)^2$,*

$$\|\theta(t) - \theta(0)\| \le u\|\theta(0)\|e^u. \tag{13}$$

*Remark* 3.5. Theorem 3.4 provides a technical condition (equation 11) under which the global convergence of continuous solutions to the DI is guaranteed, cf. equation 12, and that the solutions stay in the proximity of initialization, cf. equation 13. It is the foundation of all our subsequent results.

*Proof of Theorem 3.4.* If $\mathcal{L}(\theta(0)) = 0$, then the result holds. If $\mathcal{L}(\theta(0)) > 0$, then set $U(s) \stackrel{\text{def}}{=} \sqrt{2}\|X\|_{op}(\|W(0)\|_F + \|V(0)\|_F)s \cdot e^{\sqrt{2}\|X\|_{op}s}$ and let $G \stackrel{\text{def}}{=} B(\theta(0), 2U(2a/\alpha^2))$. By Proposition 2.1, there exists $T > 0$ and a solution $\theta \colon [0, T) \to \mathbb{R}^D$ to the DI equation 3, which can be extended up until it hits the boundary of $G$. Assume that $\theta$ is already such an extension. By Proposition 3.2, $\bar{\mathcal{L}}(t) = \int_0^t \sqrt{\mathcal{L}(\theta(s))}\, ds$ solves equation 9, whence Lemma 3.3 asserts that if $\alpha > 0$ and equation 11 is satisfied, then $\bar{\mathcal{L}}(t)$ is bounded from above by $2a/\alpha^2$ and $\bar{\mathcal{L}}'(t) = \sqrt{\mathcal{L}(\theta(t))}$ is bounded from above by $ae^{-t\alpha^2/2}$ for all $t \in [0, T)$.

By Lemma 3.1, $\|\theta(t) - \theta(0)\| \le U(\bar{\mathcal{L}}(t)) \le U(2a/\alpha^2)$ for all $t \in [0, T)$, so $\theta$ never reaches the boundary of $G$, whence $T = \infty$ and equation 12 follows. Estimating $\|W(0)\|_F + \|V(0)\|_F \le \sqrt{2}\|\theta(0)\|$ gives equation 13. □

# 4 Convergence of the Differential Inclusion Trajectories

To verify that equation 11 holds WHP at initialization, we need to impose some additional assumptions on the data matrices $X, Y$, and on the initialization scheme of $\theta_0$. In this section, all the complexity notations $\mathcal{O}, \Omega, \Theta$, etc., are understood in terms of $N$ approaching infinity, e.g., for any space $\mathcal{X}$ and a function $f \colon \mathcal{X} \times \mathbb{N} \to \mathbb{R}$, we say that $f(x, N) = \mathcal{O}(N)$ if $|f(x, N)| \leq CN$ for some constant $C > 0$ and all $x \in \mathcal{X}$.

Recall that a random variable $z \in \mathbb{R}$ is sub-Gaussian if its Orlicz norm defined as $\|z\|_{\psi_2} \overset{\text{def}}{=} \inf\{t > 0 \colon \mathbb{E}\exp(z^2/t^2) \leq 2\}$ is finite. A random vector $Z \in \mathbb{R}^n$ is said to be sub-Gaussian if $\|Z\|_{\psi_2} \overset{\text{def}}{=} \sup_{t \in \mathbb{R}^n, \|t\|_2 = 1} \|\langle Z, t\rangle\|_{\psi_2}$ is finite. For more refined treatment of the Orlicz norms and sub-Gaussian random variables, we refer the reader to Vershynin (2018).

In the sequel, we impose the following assumption.

**Assumption 4.1.**

1. $X_{i:}$'s are random i.i.d. sub-Gaussian vectors s.t. $\|X_{i:}\|_2 = \sqrt{d_0}$ and $\|X_{i:}\|_{\psi_2} = \mathcal{O}(1)$ for $i \in [N]$.

2. $(W_0)_{ij} \sim \mathcal{N}(0, \beta_w^2)$ for $(i, j) \in [d_0] \times [d_1]$ and some $\beta_w > 0$.

3. $(V_0)_{ij} \sim \mathcal{N}(0, \beta_v^2)$ for $(i, j) \in [d_1] \times [d_2]$ and some $\beta_v > 0$.

4. $W_0$ and $V_0$ are independent random matrices.

5. $\beta_w \beta_v \sqrt{d_0 d_1} = \Omega(1)$.

6. $\|Y_{i:}\|_2 = \mathcal{O}(\sqrt{d_2})$ for $i \in [N]$.

The result below provides a lower bound on $\alpha_0(0)$. The proof is a slight modification of the argument from (Nguyen et al., 2021, Theorem 5.1). Since we believe that there is a minor mistake in the proof therein (namely, in the integral estimates in Appendix E.1), we present our proof in Appendix C for completeness. We stress however that the main idea and authorship of the result is due to Nguyen et al. (2021).

**Theorem 4.2.** *Under Assumption 4.1, let $d_0 \in [N^{\delta_0}, N]$ for some $\delta_0 \in (0, 1)$. Let $\Psi \colon \mathbb{N} \to [1, \infty)$ be s.t. $d_1 \geq \max\left(N, C(\delta_0) d_0^{-1} N \Psi(N) \log^2(N)\right)$ for some $C(\delta_0) > 0$ depending on $\delta_0$ only. Then, there exists a universal constant $c(\delta_0)$ depending on $\delta_0$ only, s.t. $\alpha_0(0) \geq \sqrt{c(\delta_0) d_0 d_1} \beta_w$ holds with probability at least $1 - \exp(-\Psi(N)) - \mathcal{O}(N^2)\exp(-\Omega(N^{\delta_0/2}))$.*

The following lemma follows from standard concentration inequalities – we provide the proof for completeness in Appendix D.

**Lemma 4.3.** *If Assumption 4.1 is satisfied, then*

$$\|W_0\|_F = \Theta(\sqrt{d_0 d_1}\beta_w) \quad and \quad \|V_0\|_F = \Theta(\sqrt{d_1 d_2}\beta_v)$$

*with probability $1 - 2\exp(-\Omega(d_0 d_1)) - 2\exp(-\Omega(d_1 d_2))$,*

$$\|X\|_{op} = \mathcal{O}(\sqrt{\max\{N, d_0\}})$$

*with probability $1 - \exp(-\Omega(\max\{N, d_0\}))$, and*

$$\mathcal{L}(\theta_0) = \mathcal{O}(\|Y\|_F^2 + \beta_v^2 d_2 \|W_0\|_F^2 \|X\|_{op}^2 \log(N))$$

*with probability. $1 - \exp(-\Omega(d_2 \log(N)))$.*

Combining results from Sections 3 and 4 we obtain the following result demonstrating the global convergence of solutions to DI equation 3 towards zero loss under initialization satisfying Assumption 4.1. The full proof is provided in Appendix E.

**Corollary 4.4.** *Under Assumption 4.1, let $\beta_v^2 = d_1^{-\rho}$ for some $\rho \geq 0$ and $d_0 \in [N^{\delta_0}, N]$ for some $\delta_0 \in (0, 1)$. Let moreover $c(\delta_0)$ and $C(\delta_0)$ be as in Theorem 4.2 and*

$$d_1 \geq \max\left(N, C(\delta_0)\left[\frac{d_2 N^{2.5}}{d_0 \beta_w^2}\right]^{1/(1+\rho)} \log^2(N)\right).$$

*Then, any solution $\theta \colon [0, T] \to \mathbb{R}$ to the DI equation 3 can be extended to a solution on $\mathbb{R}_+$ and any such extension satisfies*

$$\mathcal{L}(\theta(t)) \leq \mathcal{L}(\theta(0)) \cdot \exp\left(-t \cdot c(\delta_0) d_0 d_1 \beta_w^2\right)$$

*for all $t \geq 0$ with probability at least $1 - \exp\left(-\frac{d_0}{N} \cdot \left[\frac{d_2 N^{2.5}}{d_0 \beta_w^2}\right]^{1/(\rho+1)}\right) - \mathcal{O}(N^2)\exp(-\Omega(N^{\delta_0/2})) - \exp(-\Omega(d_2 \log N))$.*

*Proof sketch.* By Theorem 3.4 and Theorem 4.2, it suffices to verify that $F(\theta(0), X, Y) = o(1)$. The last condition is verified WHP by means of Theorem 4.2 and Lemma 4.3. $\qquad\square$

*Remark* 4.5. Corollary 4.4 is a technical tool we use in subsequent sections to obtain the SGD convergence guarantees. Yet, it is still of interest, as it allows us to deduce that under the standard LeCun initialization ($\beta_w^2 = 1/d_0$, $\rho = 1$), the overparametrization bound of the order $\widetilde{\Omega}(N^{1.25})$ is sufficient for the global convergence of the continuous trajectories. We comment more on comparison with other results in Remarks 5.3 and 5.4.

## 5 Convergence of the Stochastic Gradient Descent Iterations

Let us consider a discrete version of the dynamics given by the DI Cauchy problem equation 3, i.e., the stochastic gradient descent. We start with introducing some additional notation.

Let $(\Xi, \mathcal{F}, \mu)$ be a probability space and consider a function $f \colon \mathbb{R}^D \times \Xi \to \mathbb{R}$, s.t. $f(\cdot, s)$ is locally Lipschitz for all $s \in \Xi$. Let $\theta_0 \in \mathbb{R}^D$ be a random variable with absolutely continuous distribution function. For a fixed stepsize $\eta > 0$, we say that a sequence of $\mathbb{R}^D$-valued random variables $(\theta_k^\eta)_{k \in \mathbb{N}}$ is an $f$-SGD sequence if

$$\theta_0^\eta = \theta_0; \quad \theta_{k+1}^\eta \in \theta_k^\eta - \eta \cdot \partial f(\theta_k^\eta, \xi_{k+1}) \text{ for } k \in \mathbb{N}, \tag{14}$$

where $\partial f(\theta, s)$ is the Clarke subdifferential at point $\theta$ applied to the function $\theta \mapsto f(\theta, s)$ and $(\xi_k)_{k \in \mathbb{N}_+}$ is a sequence of i.i.d. $\Xi$-valued random variables distributed according to $\mu$, which are independent of $\theta_0$.

For $b \in [N]$, let $[N]^{(b)}$ denote the family of subsets of $[N]$ containing exactly $b$ elements and $A_b \sim \text{Unif}([N]^{(b)})$ be a random variable selecting each item from $[N]^{(b)}$ with the same probability. We define the loss function $\mathcal{L}^b \colon \mathbb{R}^D \times [N]^{(b)} \to \mathbb{R}_+$ for a batch sample of size $b \in [N]$ via the formula $\mathcal{L}^b(\theta, A) \overset{\text{def}}{=} \frac{1}{2} \sum_{i \in A} \|Y_{i:} - \hat{Y}_{i:}\|^2$. Therefore, an $\mathcal{L}^b$-SGD sequence is any random sequence $(\theta_k^\eta)_{k \in \mathbb{N}}$ satisfying equation 14 with $\Xi = [N]^{(b)}$ and an i.i.d. sequence $\xi_k \sim \text{Unif}([N]^{(b)})$ for $k \in \mathbb{N}_+$. We stress that this construction corresponds to the usual mini-batch SGD.

Corollary 4.4 states that, assuming enough overparametrization, the continuous trajectories given by the dynamics of the DI problem equation 3 converge to the global minima of the loss $\mathcal{L}$ if the initial value $\theta_0$ is chosen properly, which happens WHP. In the theorem below we deduce an analogous convergence result for the $\mathcal{L}^b$-SGD iterates defined above.

**Theorem 5.1.** *Under Assumption 4.1, let $\beta_v^2 = d_1^{-\rho}$ for some $\rho > 0$ and $d_0, d_1, d_2, \delta_0, c(\delta_0), C(\delta_0)$ be as in Corollary 4.4. Choose any error $\varepsilon > 0$, batch size $b = b(N) \in [N]$ and any family $\{(\theta_k^\eta) \colon \eta > 0\}$ of $\mathcal{L}^b$-SGD sequences equation 14.*

*Then, there exists a step size $\eta_0 \in (0, 1)$ s.t. for a.e. $\eta \in (0, \eta_0)$, $\mathcal{L}(\theta_{k^*}^\eta) < \varepsilon$ for some*

$$k^* \leq \left\lfloor 1 + \frac{N}{\eta b} \max\left(0, \frac{\log\left(CN \log(N) d_0 d_1 \beta_w^2 \beta_v^2 / \varepsilon\right)}{c(\delta_0) d_0 d_1 \beta_w^2}\right)\right\rfloor, \tag{15}$$

*where $C > 0$ is some absolute constant. The result holds with probability at least $1 - \exp\left(-\frac{d_0}{N} \cdot \left[\frac{N^{2.5}}{d_0 \beta_w^2}\right]^{1/(\rho+1)}\right) - \mathcal{O}(N^2)\exp(-\Omega(N^{\delta_0/2})) - \exp(-\Omega(d_2 \log N))$.*

*Remark* 5.2. Note that $k^*$ in Theorem 5.1 depends on $\varepsilon$ via $\log(1/\varepsilon)$, i.e., SGD converges to the global minima at a linear rate.

*Remark* 5.3. In order to compare the bounds obtained by Theorem 5.1 with other works, one has to take into consideration not only parameters $\beta_w, \beta_v$ but also scaling of the data matrices $X$ and $Y$. E.g., Oymak & Soltanolkotabi (2020) works under the assumptions that $\|X_{:i}\| = 1$ for $i \in [N]$ and $\beta_w = 1$, which by the properties of Gaussian distribution corresponds exactly to our case $\|X_{:i}\| = \sqrt{d_0}$ and $\beta_w = 1/\sqrt{d_0}$.

*Remark* 5.4. Corollary 5.1 under the LeCun initialization, $\beta_w^2 = 1/d_0$, $\beta_v^2 = 1/d_1$, yields exponential loss convergence WHP for $d_1 = \tilde{\Omega}(N^{1.25})$, improving on $d_1 = \Omega(N^2)$ due to Nguyen (2021). Similarly, under different but equivalent scaling, (Oymak & Soltanolkotabi, 2020, Corollary 2.4) shows that overparametrization of the form $d_1 = \Omega(N^4/d_0^3)$ is sufficient for exponential loss convergence, when only the first layer is trained for $d_0 \in [\sqrt{N}, N]$, whereas the second layer is fixed. Neglecting the logarithmic factor, one can see that our bound $d_1 = \tilde{\Omega}(N^{1.25})$ improves upon $d_1 = \Omega(N^4/d_0^3)$ for $\delta_0 \leq 2.75/3 \approx .92$, including practical datasets dimensions. Moreover, our bound works also for $\delta_0 \in (0, 0.5)$ and for any $d_2$ (while they assume $d_2 = 1$). Finally, a simple adaptation of our technique combined with some observations from Oymak & Soltanolkotabi (2020) allows to obtain the bound $d_1 = \Omega(N^5/d_0^4)$ in training one layer setup, cf. Appendix G.

The main tool used to obtain Theorem 5.1 is the following abstract result, which claims that under some technical conditions on $f$ and initialization scheme, the solutions to the DI involving $f$ are WHP close in the supremum norm to the trajectories of the corresponding piecewise interpolated processes.

**Theorem 5.5** (Bianchi et al. (2022)). *For any probability space $(\Xi, \mathcal{F}, \mu)$, let $f : \mathbb{R}^D \times \Xi \to \mathbb{R}$ be s.t. for some function $\kappa : \mathbb{R}^D \times \Xi \to \mathbb{R}_+$, the following conditions are satisfied:*

1. $\forall\, x \in \mathbb{R}^D$, $\exists\, \varepsilon > 0$, $\forall\, z, y \in B(x, \varepsilon)$, $\forall\, s \in \Xi$, $\|f(y,s) - f(z,s)\| \leq \kappa(x,s)\|y - z\|$;

2. $\forall\, x \in \mathbb{R}^D$, $\exists\, K > 0$, $\mathbb{E}_{\xi \sim \mu}\kappa(x,\xi) \leq K(1 + \|x\|)$;

3. $\forall\, {}_{\mathcal{K} \text{ is compact}}^{\mathcal{K} \subset \mathbb{R}^D \text{ s.t.}}$, $\sup_{x \in \mathcal{K}} \mathbb{E}_{\xi \sim \mu}\kappa(x,\xi)^2 < \infty$;

4. *for a.e. $x \in \mathbb{R}^D$, $f$ is $\mathcal{C}^2$ in some neighborhood of $x$.*

*Then, for any time horizon $T > 0$, the following DI problem is well-defined*

$$\dot{\theta}(t) \in -\partial \mathbb{E}_{\xi \sim \mu} f(\theta(t), \xi) \quad \text{for a.e. } t \in [0, T]. \tag{16}$$

*Moreover, if $\{(\theta_k^\eta)_{k \in \mathbb{N}_+} : \eta > 0\}$ is a family of $f$-SGD sequences equation 14 initialized at random continuously distributed $\theta_0$, then there exists a set $\mathcal{N} \subset (0, \infty)$ s.t. $\mathcal{N}^c$ is of zero Lebesgue measure and s.t. for every compact set $\mathcal{K} \subset \mathbb{R}^D$, time horizon $T > 0$, and error $\tilde{\varepsilon} > 0$,*

$$\lim_{\mathcal{N} \ni \eta \to 0^+} \mathbb{P}\big(\exists\, \theta : [0,T] \to \mathbb{R}^D \text{ solving equation 16}, \theta(0) \in \mathcal{K}, \sup_{t \in [0,T]} |\theta(t) - \bar{\theta}^\eta(t)| < \tilde{\varepsilon} \,\big|\, \theta_0 \in \mathcal{K}\big) = 1,$$

*where $\bar{\theta}^\eta$ is the corresponding random (measurable w.r.t. $(\theta_k^\eta)_{k \in \mathbb{N}}$) piecewise interpolated process defined, i.e.,*

$$\bar{\theta}^\eta(t) \stackrel{\text{def}}{=} \theta_k^\eta + (t/\eta - k)(\theta_{k+1}^\eta - \theta_k^\eta) \tag{17}$$

*for all $t \in [k\eta, (k+1)\eta), k \in \mathbb{N}$.*

The following theorem built upon Bianchi et al. (2022) can be seen as a general tool allowing to pass (when deducing global convergence) from the solutions to the DI equation 3 to the SGD sequences given by equation 14. We state it for general approximators (including, e.g., deep ReLU NN) and general loss functions as we believe it is of independent interest. In particular, we drop the assumption on the MSE loss and the NN denoted by $\hat{Y}$.

**Theorem 5.6.** *Let $\tilde{\mathcal{L}}_i : \mathbb{R}^D \to \mathbb{R}$ for $i \in [N]$ be arbitrary locally Lipschitz functions satisfying the chain rule equation 2 and being $\mathcal{C}^2$ in some neighborhood of a.e. point of $\mathbb{R}^D$. Set $\tilde{\mathcal{L}} = \sum_{i \in [N]} \tilde{\mathcal{L}}_i$. Assume there exists a nonempty compact sets $Q \subset G \subset \mathbb{R}^D$, s.t. any solution $\theta : [0, \infty) \to \mathbb{R}^D$ to the DI*

$$\dot{\theta}(t) \in -\partial \tilde{\mathcal{L}}(\theta(t)) \quad \forall\, t \geq 0, \tag{18}$$

*if initialized in $Q$, remains in $G$ and satisfies $\tilde{\mathcal{L}}(\theta(t)) \leq \tilde{\mathcal{L}}(\theta(0))e^{-\gamma t}$ for all $t \geq 0$ and some $\gamma > 0$. Choose confidence threshold $\delta > 0$, error $\varepsilon > 0$, batch size $b \in [N]$, and family $\{(\theta_k^\eta)_{k \in \mathbb{N}} \colon \eta > 0\}$ of $\tilde{\mathcal{L}}^b$-SGD sequences given by equation 14, where $\Xi = [N]^b$, $\mu = \mathrm{Unif}([N]^b)$ and $\tilde{\mathcal{L}}^b \colon \mathbb{R}^D \times [N]^b \to \mathbb{R}_+$ is given by $\tilde{\mathcal{L}}^b(\theta, A) = \sum_{i \in A} \tilde{\mathcal{L}}_i(\theta)$. Assume that $\theta_0$ is continuously distributed.*

*Then, there exists a step size $\eta_0 \in (0,1)$ s.t. for a.e. $\eta \in (0, \eta_0)$, $\mathbb{P}(\tilde{\mathcal{L}}(\theta_{k^*}^\eta) < \varepsilon \,|\, \theta_0 \in Q) \geq 1 - \delta$ for $k^* \leq \lfloor 1 + \frac{N}{\eta b} \max(0, \gamma^{-1} \log(2\varepsilon^{-1} \sup_{\theta \in Q} \tilde{\mathcal{L}}(\theta))) \rfloor$.*

*Proof sketch of Theorem 5.6.* Let $l \overset{\text{def}}{=} \sup_{\theta \in Q} \tilde{\mathcal{L}}(\theta)$ and $T^* \overset{\text{def}}{=} \inf\{\, t \geq 0 \colon le^{-\gamma t} \leq \varepsilon/2 \,\} = \max\left(0, \frac{\log(2l/\varepsilon)}{\gamma}\right)$ so that all solutions to the DI equation 18 initialized in the set $Q$ fall to $\tilde{\mathcal{L}}^{-1}([0, \varepsilon/2])$ before time $T^*$ (and clearly never escape it). Set $L \overset{\text{def}}{=} \sup\{\, \|v\| \colon v \in \partial L(\theta), \theta \in G \,\}$.

If we could apply Theorem 5.5 with the family $\{(\theta_k^\eta)_{k \in \mathbb{N}} \colon \eta > 0\}$, $\tilde{\varepsilon} = \min(\varepsilon/2L, 1)$ and $T = 1 + \frac{N}{b}T^*$, it would yield that for any $\delta \in (0, 1)$, there exists $\eta_0 \in (0, 1)$ s.t. for a.e. $\eta \in (0, \eta_0)$,

$$\mathbb{P}\big(\exists\, \theta \text{ solving } \dot{\theta}(t) \in -\partial\mathbb{E}\tilde{\mathcal{L}}^b(\theta(t), A_b) \text{ s.t. } \theta(0) \in Q \text{ and } \sup_{t \in [0,T]} |\theta(t) - \bar{\theta}^\eta(t)| < \tilde{\varepsilon} \,\big|\, \theta_0 \in Q\big) \geq 1 - \delta.$$

Recall that $A_b \sim \mathrm{Unif}([N]^b)$ and note that $\mathbb{E}\tilde{\mathcal{L}}^b(\cdot, A_b) = \frac{b}{N}\tilde{\mathcal{L}}(\cdot)$, whence if $\theta(t)$ solves $\dot{\theta}(t) \in -\partial\mathbb{E}\tilde{\mathcal{L}}^b(\theta(t), A_b)$, then $\theta(tN/b)$ solves equation 18. In particular $\tilde{\mathcal{L}}(\theta(t)) \leq \varepsilon/2$ for any $t \geq \frac{N}{b}T^*$. Therefore, as for $\eta \in (0, \eta_0)$ it holds that $\frac{N}{b}T^* \leq \eta k^* \leq T$, then for a.e. $\eta \in (0, \eta_0)$,

$$\tilde{\mathcal{L}}(\theta_{k^*}^\eta) = \tilde{\mathcal{L}}(\bar{\theta}^\eta(\eta k^*)) \leq \tilde{\mathcal{L}}(\theta(\eta k^*)) + \tilde{\varepsilon}L \leq \varepsilon$$

with probability at least $1 - \delta$ conditioned on $\theta_0 \in Q$.

However, in general $\tilde{\mathcal{L}}$ does not satisfy the assumptions of Theorem 5.5. In order to overcome this, we need to consider the set $G$ and modify $\tilde{\mathcal{L}}$ outside of some neighborhood containing $G$, so that it becomes globally Lipschitz. As all solutions to equation 18 initialized in $Q$ remain in $G$, then it turns out that such modification does not conflict with the argument above, as is discussed in detail in Appendix F. $\qquad\square$

We are ready to prove the main result of this section.

*Proof of Theorem 5.1.* For $\theta = (W, V) \in \mathbb{R}^D$ and $\tilde{X} \in \mathbb{R}^{N \times d_0}$, let $\alpha_0(\tilde{X}, \theta) \overset{\text{def}}{=} \sigma_{min}(\phi(\tilde{X}W)^T)$ and $\mathcal{L}(\tilde{X}, \theta) \overset{\text{def}}{=} \frac{1}{2}\|Y - \phi(\tilde{X}W)V\|_F^2$. Define

$$Q(\tilde{X}) \overset{\text{def}}{=} \{\, \theta \in \mathbb{R}^D \colon F(\theta, \tilde{X}, Y) < \tfrac{1}{8}, \quad \alpha_0(\tilde{X}, \theta) \geq \sqrt{c(\delta_0)d_0 d_1}\beta_w,$$
$$\mathcal{L}(\tilde{X}, \theta) \leq Cd_0 d_1 d_2 \beta_w^2 \beta_v^2 N \log(N), \quad \|\theta\| \leq C(\sqrt{d_0 d_1}\beta_w + \sqrt{d_1 d_2}\beta_v)\},$$

where $c(\delta_0)$ is the same constant as in Theorem 4.2, $F$ is defined as in Theorem 3.4, equation 11, and $C > 0$ is some big enough absolute constant such that

$$\mathbb{P}(\theta_0 \in Q(X)) \geq 1 - \exp\big(-\frac{d_0}{N} \cdot \Big[\frac{N^{2.5}}{d_0\beta_w^2}\Big]^{1/(\rho+1)}\big) - \mathcal{O}(N^2)\exp(-\Omega(N^{\delta_0/2})) - \exp(-\Omega(d_2 \log N)),$$

which is possible in virtue of Theorem 4.2 and Lemma 4.3, cf. Proof of Corollary 4.4. For each $\tilde{X}$, let $u \overset{\text{def}}{=} u(\tilde{X}, \theta) = 2\sqrt{\mathcal{L}(\tilde{X}, \theta)}/\alpha_0^2(\tilde{X}, \theta)$ and $U(\tilde{X}) \overset{\text{def}}{=} \sup_{\theta \in Q(\tilde{X})}\{\, \sqrt{2}\|\tilde{X}\|_{op}\|\theta\|u \cdot e^{\sqrt{2}\|\tilde{X}\|_{op}u} \,\}$, so that any solution to the DI $\dot{\theta} \in -\partial\mathcal{L}(\theta)$, if initialized in $Q(\tilde{X})$, remains in the set $G(\tilde{X}) = B(Q(\tilde{X}), U(\tilde{X}))$ in virtue of Lemma 3.1 (cf., Proof of Theorem 3.4). Moreover, $U(\tilde{X}) < \infty$ by compactness of $Q(\tilde{X})$, whence $G(\tilde{X})$ is compact.

For each $\tilde{X}$, apply Theorem 5.6 with $\tilde{\mathcal{L}}(\cdot) = \mathcal{L}(\tilde{X}, \cdot)$, $Q = Q(\tilde{X})$, $\gamma = \inf_{\theta \in Q(\tilde{X})}\alpha_0^2(\tilde{X}, \theta)$, $\delta = \mathbb{P}(\theta_0 \notin Q(\tilde{X}))$ and $G = G(\tilde{X})$, to get that for some $\eta_0 \in (0, 1)$ and a.e. $\eta \in (0, \eta_0)$, $\mathbb{P}(\mathcal{L}(\tilde{X}, \theta_{k^*}^\eta) < \varepsilon \,|\, \theta_0 \in Q(\tilde{X})) \geq \mathbb{P}(\theta_0 \in Q(\tilde{X}))$, where $k^*$ is as in Theorem 5.6 and whence bounded as in equation 15 by the definition of $Q(\tilde{X})$. Note that $\eta_0$ depends on $\tilde{X}$ only.

Using the inequality $\mathbb{P}(A \,|\, B) \leq \mathbb{P}(A)/\mathbb{P}(B)$, multiplying both sides by $\delta$, integrating w.r.t. the distribution of $X$ and estimating $(1 - \delta)^2 \geq 1 - 2\delta$, we get that $\mathbb{P}(\exists\, \eta_0 \in (0,1)$ s.t. for a.e. $\eta \in (0, \eta_0)$, $\mathcal{L}(\theta_{k^*}^\eta) < \varepsilon)$ is at least $1 - 2\mathbb{P}(\theta_0 \notin Q(X))$, as desired. $\qquad\square$

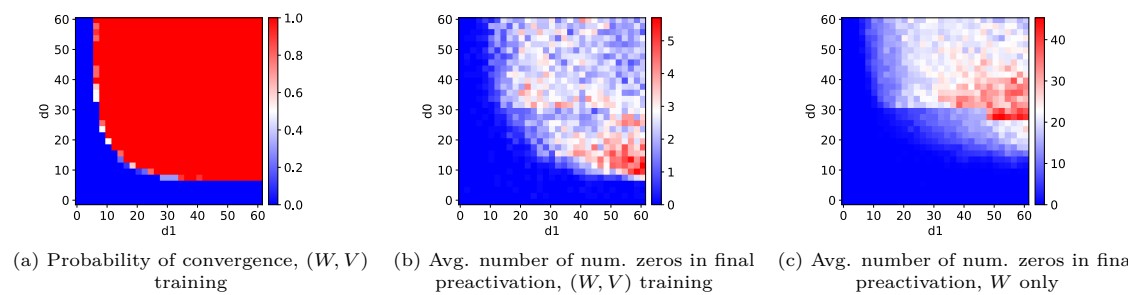

(a) Probability of convergence, $(W, V)$ training

(b) Avg. number of num. zeros in final preactivation, $(W, V)$ training

(c) Avg. number of num. zeros in final preactivation, $W$ only

Figure 2: Numerical results for both training setups after $50k$ SGD iterates.

# 6 Numerical Experiments

We present some numerical results illustrating two training setups – when both layers $(W, V)$ are trained and when $W$ is trained only. As shown in (Oymak & Soltanolkotabi, 2020, Section 4), it is expected that overparametrization of the order $\mathcal{O}(N/d_0)$ is sufficient for global convergence of SGD iterations in case when only the hidden layer is trained. We complement their findings by analyzing the two layer training regime, observing that a similar phenomenon occurs empirically, thus showing that some further investigation of the optimization of shallow neural networks (beyond our Theorem 5.1) is still required. We also compare with a recent work Elkabetz & Cohen (2021), where the authors took approach similar to ours, i.e., they start with analysis of continuous solutions and pass from there to SGD iterates. We show that, in contrast to ours, in case of ReLU networks their approach based on Hessian analysis along the trajectories cannot be applied, as these cross the boundaries of differentiability regions. Some additional findings are also reported.

## 6.1 Setup

Data is generated per single experimental run as follows: $N = 200$, rows of $X$ are i.i.d. from the unit sphere, $d_2 = 1$ and labels $Y$ are randomly chosen s.t. half are set to 1 and the other half to $-1$. In the first training setup $W$ has i.i.d. $\mathcal{N}(0, 1)$ entries and $V$ has i.i.d. $\mathcal{N}(0, 1/d_1)$ entries. In the second training setup $W$ is as before and $V$ is fixed – half of the entries are $1/\sqrt{d_1}$ and half are $-1/\sqrt{d_1}$ as in Oymak & Soltanolkotabi (2020). In all of the experiments we vary $d_0, d_1$. The NNs are implemented within the Pytorch framework. We used the standard SGD optimizer (in fact, GD as the batch size is set to 200) with momentum (0.9). The learning rate differs on the training setup and is set to 0.15 ($W$ only training), or 0.002 ($(W, V)$ training).

## 6.2 Results

Figure 2a illustrates the probability of convergence towards a global minimum depending on the network configuration. The probability is approximated based on 10 independent runs and $d_0, d_1$ grid 2 spaced, the convergence criterion is $\|\hat{y} - y\| / \|y\| < 2.5 \times 10^{-3}$ as in Oymak & Soltanolkotabi (2020). Compared with Oymak & Soltanolkotabi (2020), there seems to be no difference between training setups in terms of convergence probability and it is supposed that the overparametrization $\mathcal{O}(N/d_0)$ is sufficient for the global SGD convergence. In Figures 2b, 2c we present the average number of numerical zeros (absolute values below $10^{-8}$) in the preactivation layer at convergence. Our investigation reveals an SGD optimization bias in both setups toward global minima with positive number of zero preactivation neurons (i.e., ReLU non differentiability points). In fact, these seem to be points of intersection of several ReLU activation pattern regions, as there are many zeros found. Note the different scales of the two plots – the $W$ only training setup results in order of magnitude more numerical zeros than in the case of $(W, V)$ training. This in particular suggests that the training trajectories might cross many different ReLU regions and thus they would be far from the linear regime described in Elkabetz & Cohen (2021). Below, we investigate further this phenomenon.

We now turn to Figures 3 in which we analyze the training trajectories for both setups. It is seen that despite being close to global minima (loss is already close to 0 as seen on Figures 3a, 3f), the number of numerical zeros in the preactivation pattern stays positive and is confined to a small range of values depending on the studied overparametrization level as presented on Figures 3b, 3g. This confirms the observation above that

the GD scheme prefers minima located close to the boundaries between several ReLU activation patterns. In fact, these seem to be corner points connecting several regions. We are not aware of any explanation of such a phenomenon in the literature. Moreover, despite being close to global minima, the activation patterns keep changing while performing the consecutive GD iterates before eventually stabilizing in some region. At which iteration that happens, depends on the overparametrization level as presented on Figures 3c, 3h. This, in particular, demonstrates that most of the shallow ReLU networks training scheme happens in the nonlinear regime, i.e., it is not confined to a single ReLU activation region until the very end stage of training. The activation regions keep changing in a nonlinear fashion. Hence, the problem of studying the convergence of ReLU nets cannot be simplified to a study within a linear regime as suggested by Elkabetz & Cohen (2021).

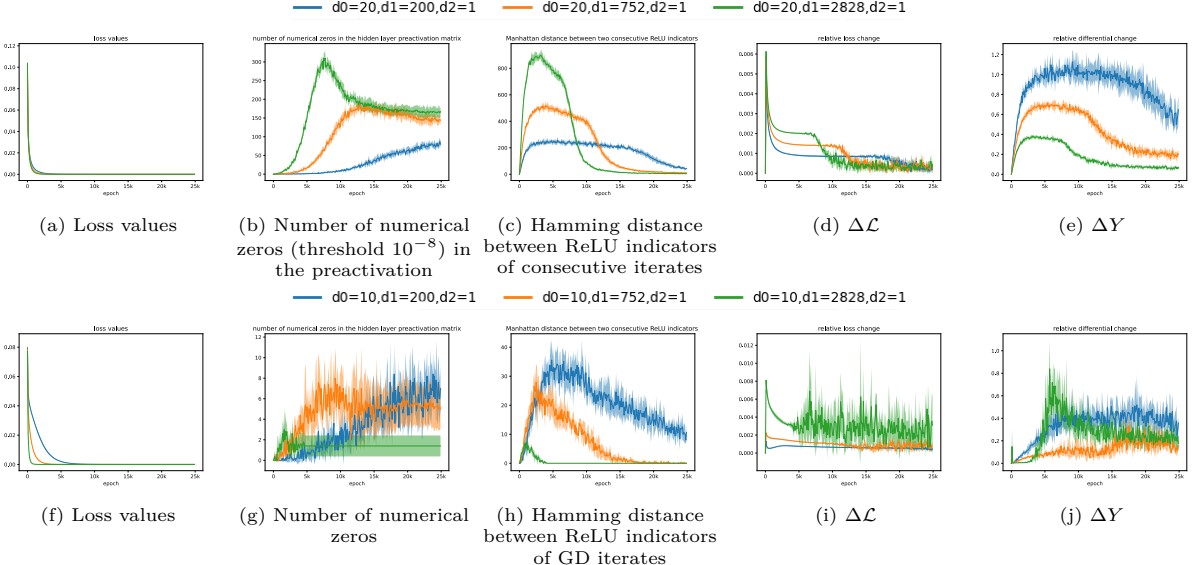

Figure 3: Top row: training the hidden layer of the shallow NN only. Bottom row: training both layers of the shallow NN. The number of the hidden neurons is varied (the NN configuration is provided in the legend) and the total number of epochs of the evolution is equal to $25k$. The solid curve presents the mean from five independent runs, and the shaded region presents the standard deviation, plotted every 100th epoch. In Figures 3b,3g by the number of numerical zeros we mean the count of absolute values smaller than the given threshold in the hidden layer preactivation matrix (before applying ReLU).

Finally, on Figures 3d, 3i, 3e, 3j we investigated the relative loss change $\Delta\mathcal{L} = \frac{|\mathcal{L}(\theta_k) - \mathcal{L}(\theta_{k-1})|}{\mathcal{L}(\theta_{k-1})}$ and the relative differential change measured in the operator norm $\Delta D = \frac{\|D\hat{Y}_k - D\hat{Y}_{k-1}\|_{op}}{\|D\hat{Y}_{k-1}\|_{op}}$, where $D\hat{Y}_k$ is the differential of $\hat{Y}$ (NN output) with respect to $\theta_k$ (NN weights vector). It is visible that the relative differential change is by order of magnitude larger than the relative loss change, suggesting that the training for moderate and larger overparametrization levels is far from the lazy training regime studied in Chizat et al. (2019). Where the general criterion for the lazy training regime is given by $\Delta\mathcal{L} \gg \Delta D$, cf. (Chizat et al., 2019, Section 1.2).

## 7 Conclusions and Future Work

We have demonstrated an improved trainability overparametrization bound of order $\tilde{\Omega}(N^{1.25})$ on the hidden layer of shallow NN equipped with ReLU activation functions. We have obtained Theorem 5.6 – an result allowing to pass from continuous solutions of the DI to the dynamics of SGD. We believe that our contribution deepens the understanding of the optimization theory of NN. There are several natural directions of further research and we list some of them below. First direction is towards the theory of deep networks, where one could try to combine Theorem 5.6 with an analysis of DI dynamics in order to obtain improved overparametrization guarantees. Secondly, Theorem 5.6 might serve as a tool to obtain overparametrization bounds which are suggested by numerical experiments in Section 6. Finally, all known bounds for ReLU NNs are valid under strong, probabilistic data assumptions and it would be of interest to pursue directions of research that would allow for more general data such as in the case of smooth activations, cf. Table 1.

## 8 Acknowledgements

BP and JC were partially supported by NAWA Polish Returns grant PPN/PPO/2018/1/00029.

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
