# OpenReview forum: "Improved Overparametrization Bounds for Global Convergence of SGD for Shallow Neural Networks"
_TMLR — Accepted by TMLR_

### Review · Reviewer_7vWr · 2022-12-07

**Summary Of Contributions:**

The paper studies sufficient bounds on over-parameterization for achieving vanishing training loss for two-layer ReLU networks trained via (stochastic) gradient descent on arbitrary data satisfying certain regularity conditions.
The obtained bounds improve on previous works which considered the ReLU, in particular Nguyen (2021) and Oymak & Soltanolkotabi (2020). Importantly, the result relies on a continuous-time, non-smooth analysis of the differential inclusion analogous to the gradient flow, which avoids discrete time analyses which seemed to incur additional need for over-parameterization in previous works.
The continuous-time results are extended to discrete-time through appropriate mini-batch SGD discretization, adapting results from Bianchi et al. (2022), and experiments are provided to illustrate the findings.

**Audience:**

Yes

**Claims And Evidence:**

Yes

**Requested Changes:**

* it would be helpful to provide more insight on what the step-size needs to be in order for the SGD results to apply, if that's available, or at least comment on this issue
* Theorem 1.1/1.2 are a bit hard to parse without reading what follows, could you define the setting and architecture used more clearly before stating them?
* some comments would be welcome after Corollary 4.5, since that's the first place where the over-parameterization bounds appear. In particular, perhaps some of the comments in Remark 5.2/3 could be moved there? Similarly, a brief discussion after Theorem 3.4 would be a helpful addition.
* legends are difficult to read in Figure 3, please increase the font or make the plots bigger.
* could you define more clearly what you mean by "number of zeros in final preactivations"? are these the number of $(i,j)$ such that $w_j^\top x_i \approx 0$?
* what is $D Y$ in the definition of $\Delta D$? Could you explain further why the proposed characterization of lazy vs active training ($\Delta L \gg \Delta D$) is valid, or where it comes from?

**Strengths And Weaknesses:**

Strenghts: improved over-parameterization requirements by leveraging a continuous-time analysis

Weaknesses:
One missing aspect is a characterization of the step-size needed for SGD to succeed. Currently Theorem 5.6 is of the flavor "there exists a step-size $\eta_0$, due to the asymptotic nature of Theorem 5.5, so it is unclear how small $\eta_0$ needs to be for optimization to succeed.

---

> ### Author Response · Authors · 2022-12-10
> **Response to Reviewer 7vWr**
>
> We appreciate very much your feedback, thank you! Please find below our response to the raised issues.
>
> **Lack of quantitative bound on the step size** -- we acknowledge this shortcoming and discuss this issue at the end of Section 1.2 (in SGD step size paragraph). In principle, in all previous results that we are aware of, obtained bound was of the form O(1/N^2), which is still far from learning rates used in ML practice. It is not clear how to obtain such a bound via our setting as we pass to the discrete case via the continuous approach, contrary to the previous works. We have added citations to the relevant results in the paper.
>
> **Setting and architecture presentation** -- we have improved on the presentation in Section 1.2 as suggested. Please let us know in case some further explanation is required.
>
> **Comments to Theorem 3.4 and Corollary 4.5** -- we have added appropriate comment providing additional insights on the meaning and importance of the above results.
>
> **Plot legends** -- we have included legends (separate) with increased fonts as requested.
>
>
> **Number of zeros in the final preactivations** -- we have added an additional explanation in the caption to Figure 3.
>
>
> **Lazy training regime explanation** -- we have added the definition of $DY$. We use the criterion for the lazy training regime  introduced in Chizat et al. (2019) "Sec.1.2 When does lazy training occur?".

---

### Review · Reviewer_fTpx · 2023-01-17

**Summary Of Contributions:**

This paper studies what width is required for an SGD-trained neural network with 1 hidden layer and ReLU activations to fit the training data. The assumption on the data is that it is drawn from a subgaussian distribution. The result is that a width of d_1 = O(N^{1.25}) on the hidden layer is sufficient to memorize the labels for N training examples.

The proof proceeds by: (1) in sections 3 and 4, studying the gradient flow solutions and proving that they have to be bounded in a compact set. (2) using the analysis of gradient flow solutions to compare to SGD using a theorem of Bianchi et al. 2022 for comparing gradient flow solutions to SGD sequences.


**Audience:**

No

**Claims And Evidence:**

Yes

**Requested Changes:**



* In this sentence: “Yet, practice suggests that with enough overparametrization, SGD iterations achieve global minima most of the times”. By global minima do you mean just that the training data is fit?

* In this phrase: “which is better for d_0 in a small neighborhood of N", do you mean, "which is better for d_0" larger than N^{2/3}?"

* Please streamline this sentence, since it is confusing: "However, existing quantitative results for ReLU NNs give to the best of our knowledge no better bound than O(1/N^2), which is still far from the learning rates used in ML practice".

* Appendix B, equation (23), last line should be an inequality

* Between equations (8) and (9), missing a parenthesis in \sqrt{\cL(\theta(s)}. Parenthesis also missing in first line of proof of Proposition 3.2.

*  In Figure 1, please add comparison to the theoretical bound, since the caption mentions it but it can't be seen in the figure. Or at least justify "Observe that empirically the upper bound is smaller than the derived theoretical bound." without having the reader do the calculation.

* The argument in Lemma 3.3 is hard to read. Please simplify it, especially considering it is in the main text. It took a minute for me to parse why the last line is a contradiction, because the exponent of alpha in the denominator at the end is different from the one in (10), which threw me off. Also, in equation (11) in the statement of Theorem 3.4, the same expression as (10) appears, but with different constant factors. These small changes are distracting and it would be better if, e.g., you just stated (10) as you state (11) and just called on (10) in the statement of Theorem 3.4.

* In Assumption 4.1, "W0 and V0 are independent random vectors.", these are matrices, right?

* In Assumption 4.1, please instead make \|Y_{i:}\|_2 be O(\sqrt{d_2}) at initialization, so that this holds without assuming Le Cun initialization. You can immediately switch to LeCun initialization if you'd like and in the proof you can use that \|Y_{i:}\|_2 = O(\beta_w \beta_v \sqrt{d_0d_1d_2}), but this looks very strange as an assumption.

* Theorem 4.3 typo, "for some and C(\delta_0)".  Appendix C typo, "We not turn to the", "for for". Appendix E typo, "tha fact"

* Equation (14) should be \theta_{k+1}^{\eta} \in \theta_k^{\eta} - \eta \cdot \partial f(\theta_k^{\eta}, \xi_{k+1}) ?

**Strengths And Weaknesses:**

The math in the paper seems correct, and the problem studied is of interest to the community.
However, I have three main concerns:

1. The literature review misses some relevant work by Daniely on "memorization with (almost) no overhead", see https://arxiv.org/pdf/1911.09873.pdf and https://arxiv.org/pdf/2003.12895.pdf . In the first of these two papers, the data points x are assumed to be drawn from a Gaussian distribution, and memorization of SGD-trained networks is proved with d_1 = O(N / d_0), which is a better rate than the one in this paper. Since the assumptions are similar in this paper (the main difference being Gaussian x vs. sub-Gaussian x), the authors should discuss the differences between their result and proof ideas and those in Daniely.

2. I am having difficulty understanding what are the new elements in this analysis. The main elements of this analysis are in Section 3, 4, 5. But it seems like the arguments in Sections 4 & 5 may already be known. The authors should be more explicit about what has been done before vs. what is new.

a. It seems that Section 4 follows from a known lower bound on the minimum eigenvalue of the NTK at initialization via general lemmas shown in Nguyen et al. 2021.

These lemmas are in the appendix as Lemma C.1 and Lemma C.3 and are re-proved, although it not clear why since they already appear in Nguyen et al. 2021. In the main text it is also stated that the proof is a slight modification of the argument of Nguyen et al. 2021. Could the authors please be more explicit in explaining the differences with that work?

Finally, the last ingredient of Appendix C is Lemma C.2, which lower-bounds the magnitudes of the Hermite coefficients of the ReLU activation function. This lemma follows from the explicit formula for the Hermite coefficients of the ReLU. See Appendix A.2 of https://arxiv.org/pdf/1910.04970.pdf, or Lemma 11 and page 18 of of https://arxiv.org/pdf/1602.05897.pdf .

b. It seems that Section 5 is also a direct application of a result of Bianchi et al. 2022.

3. The presentation of the result has many technical details in the main text. It would help the reader for them to be put in the appendix, so that in the main text the authors could focus on the key elements of the proof.

---

> ### Author Response · Authors · 2023-02-02
> **Response to Reviewer fTpx**
>
> We are grateful for the detailed review. We have applied all the requested changes in the newest revision. Please see below our response to the raised issues.
>
> **1. Comparison with Daniely et al.** -- We were unaware of these works, and thank you for pointing them out. We have added the appropriate commentary in the main body. In short:
> - The first mentioned work (https://arxiv.org/pdf/1911.09873.pdf) improves upon Oymak--Soltanolkotabi but requires weights initialization that is far from standard, making the optimization process equivalent to NTK optimization. In particular, the Author assumes that $W=[W',W']$ for some Gaussian orthogonal matrix $W'$ and $V=(B,B,\ldots,B,-B,-B,\ldots,-B)$ for some $B$ big enough. We use the standard LeCun initialization scheme but the obtained overparametrization bound is slightly off.
> - The second mentioned work (https://arxiv.org/pdf/2003.12895.pdf) works for LeCun initialization and some non-differentiable activations but is not applicable to ReLU activation (due to the assumption that $\mathbb{E}_{X\sim\mathcal{N}{(0,1)}}\sigma'(X)=0$) in contrast to the results presented in our paper.
>
> **2.a Results from Section 4** - there are three reasons for repeating the argument leading to our Theorem 4.3 (Theorem 5.1 in Nguyen (https://arxiv.org/pdf/2012.11654.pdf)).
> 1. The formulation of Theorem 5.1 is not clear in case of shallow networks (for L=2, the condition in equation (28) does not parse).
> 2. We believe that there is a slight mistake in the calculations on page 30 in the last equality in the sequence of integral estimations, namely it is not true that
> $$
>     \\mathbb{P}(
>     \\|\sigma(F\_{k-1}w)\\|\_2
>     1\_{\\|\\sigma(F\_{k-1}w)\\|\_2 > t} > \\sqrt{s} ) =
>     \\mathbb{P}(
>     \\|\\sigma(F\_{k-1}w)\\|\_2 > t
>     )
>     \\mathbb{P}(
>     \\|\\sigma(F\_{k-1}w)\\|\_2 > \\sqrt{s}
>     )
> $$
> 3. In order to make comparisons with other works easier, we decided upon making the result for initialization with arbitrary variance parameters ($\beta_w$ and $\beta_v$ in our paper).
>
> We therefore decided to repeat the whole argument for completeness, stressing many times that the authorship of the results is due to Nguyen. We decided to stress this point even more in the new revision of our work to avoid ambiguity.
>
> Finally, we thank for the reference to the explicit formula for the Hermite coefficients of the ReLU function.
>
> **2.b Results from Section 5** -
> Results in Section 5. are implied by the result of Bianchi et a., yet some additional effort is required to achieve that for the reasons explained in the Proof sketch of Theorem 5.6. in the paper. For the details, refer to Appendix F in our work.
>
> **3. Too many technical details in the main text** -- We have tried to balance the ratio of technical to non-technical matter in the article and incorporated the required changes (e.g., making the proof of Lemma 3.3 clearer). We followed a more journal-oriented style rather than a conference-oriented style with proof sketches given in the main part. However, if some particular parts require moving to the appendix, please let us know.

---

> > ### Comment · Reviewer_fTpx · 2023-02-21
> > **Response**
> >
> > I would like to thank the authors for their replies and for their edits to the paper.
> >
> > ### 1. Comparison with Daniely
> > * Really this should be highlighted/stressed more in the main text since it is the closest paper and solves the problem posed in this paper with a few small differences: 1) Daniely 2019 and 2020 are for Gaussian data instead of subgaussian, and 2) Daniely 2019 uses a nonstandard initialization, and 3) Daniely 2020 does not apply to ReLU.
> >
> > * The revision cites the wrong paper: it should cite Daniely 2019 (https://arxiv.org/pdf/1911.09873.pdf) instead of the 2016 paper by Daniely et al.
> > * The Daniely 2019 paper gives a $O(N / d_0)$ rate, and even though the initialization is "non-standard" it should appear in Table 1, since it is the best rate.
> > * The text in section 1.3 should also correctly cite the Daniely 2019 paper as achieving $\tilde{O}(N / d_0)$ rate instead of $O(N^2 / d_0)$.
> > * The Daniely 2020 paper should also appear in the table as achieving a $O(N \log(d_0)^4 / d_0)$ rate.
> >
> > ###  2. a. Results from Section 4
> > It is fine to repeat the argument of Nguyen et al. 2021, since you make clear that the authorship is theirs. In response to your comments,
> >
> > 1. See top of page 10 of Nguyen et al. 2021 for discussion on case L = 2, which implies the result applies to L = 2, and that the restrictions in (19) and (28) should be ignored since they only make sense as restrictions for L > 2. However, I do agree that it could be more clearly written.
> >
> > 2. The calculation mistake on page 30 of Nguyen et al. 2021 has a very short fix:
> > $$P[||\sigma(F_{k-1} w)|| 1(||\sigma(F_{k-1}w)||>t)>\sqrt{s}] \leq P[||\sigma(F_{k-1} w)|| > \frac{\sqrt{s} + t}{2}]$$
> > and then you continue with the bound with a slightly different constant factor.
> >
> > 3. OK.
> >
> > ### 2. b. Results from Section 5
> > OK, good point.
> >
> > ### 3. Technical details in main text
> > Good point that this is the more journal-oriented style, so it is okay. Thank you for your edits to Lemma 3.3 and Theorem 3.4.

---

> > > ### Author Response · Authors · 2023-02-22
> > > **Response**
> > >
> > > Thank you one more time for a detailed feedback.
> > >
> > > **1. Comparison with Daniely et al.**
> > >
> > > We have added both Daniely's papers to Table 1. as asked (note that $\widetilde{\Omega}$ hides the logarithmic factors).
> > > We have complemented it with additional explanation in section 1.3 and a footnote with a (rough) description of the initialization scheme from Daniely, 2019.
> > > Please let us know in case something else still requires clarification or needs to be stressed.
> > >
> > > **2.a Results from Section 4**
> > >
> > > 1. Thank you for pointing that out, we have missed that part.
> > > 2. We agree that the fix is simple (we solve this issue similarly in our proof). Still, we wanted to make it explicit to avoid ambiguity which would emerge with clarifications referring to particular lines of the proof of (a version of) the paper of Nguyen.

---

> > > > ### Comment · Reviewer_fTpx · 2023-02-22
> > > > **Response**
> > > >
> > > > ### Re: Comparison with Daniely et al.
> > > > Thank you for your edits. I made a mistake when I said that Daniely 2020a was for Gaussian data. It is for data uniform on sphere -- the table should be updated.
> > > >
> > > > ### Re: results from section 4
> > > > Sounds good.

---

> > > > > ### Comment · Reviewer_fTpx · 2023-02-22
> > > > > **Response**
> > > > >
> > > > > This may be a nitpick, but in Table 1 doesn't it make more sense to write $\tilde{O}$ instead of $\tilde{\Omega}$? Because you are claiming that $\tilde{O}(N^{1.25})$ width is sufficient to memorize $N$ examples.

---

> > > > > > ### Author Response · Authors · 2023-02-22
> > > > > > **Response**
> > > > > >
> > > > > > We have corrected the distribution as indicated.
> > > > > >
> > > > > > Regarding the notation $O$ vs $\Omega$ -- the $\Omega$ convention means that any overparametrization of this order suffices for memorization. It traces back to the seminal works Du et al., 2019 and Allen-Zhu et al., 2019 and thus we decided to embrace it. We agree that the notation $O$ might be slightly more appropriate although we see this issue more as a matter of taste and prefer to stick with $\Omega$ notation since 1) it is still popular, cf. Nguyen, 2021, 2) changing the convention throughout the whole manuscript would be a very error-prone operation.

---

> > > > > > > ### Comment · Reviewer_fTpx · 2023-02-22
> > > > > > > **Response**
> > > > > > >
> > > > > > > That makes sense, thanks.

---

### Review · Reviewer_Uxfn · 2023-02-02

**Summary Of Contributions:**

This paper improves existing asymptotic bounds on the extent of overparameterization required to ensure global convergence for single hidden layer relu networks trained with SGD. In the continuous time case, the authors formulate the training process as a differential inclusion with the time derivative of the parameters as the sub-gradient of the population loss function. In this case, a convergence rate that improves with width is also provided. The authors obtain these bounds by first analyzing the dynamics of the differential inclusion and then extending their results to the discrete time case by proving a theorem (5.5) that allows to the authors to bound the sup norm between the continuous time trajectory and the discrete time trajectory. Ultimately, the authors obtain a scaling that the width of the hidden layer should be $d_1 =  \tilde{\Omega}(N^{1.25})$ which is an improvement over more general bounds and even other bounds that focus on single hidden layer relu. There are a number of numerical results presented: I was not entirely clear on the intent or meaning of these, as discussed below.

**Audience:**

Yes

**Broader Impact Concerns:**

None.

**Claims And Evidence:**

Yes

**Requested Changes:**

I think in order for this paper to be presented clearly to a broad audience, slightly more background on differential inclusions could be added. At least some statements of what a "solution" to a DI problem means would be helpful to many readers, I believe. I am not suggesting that the authors write a separate primer on DIs.

I am not sure why there's not an attempt to study the scaling results numerically. The current set of experiments is not clearly presented, so if these are really informed directly by the analysis, then it would be nice to have a much more transparent assessment of their utility.

 Minor, but the notation $2.5e - 03$ should really be $2.5\times 10^{-3}$ for clarity.

**Strengths And Weaknesses:**

The analysis in this paper is exceedingly clear. Many of the results follow from straightforward arguments presented in the appendix and some are short enough to prove in the text. Given the strength of the results, I think this is a real accomplishment. One thing I believe could be improved (though, in part, this is due to my own deficiency) is the discussion of the implications of using a differential inclusion as opposed to an ODE for the parameters or a gradient flow (as in the case of the infinite width limit).

The biggest weakness from my perspective is the numerical experiments. In fact, I would be satisfied with this paper if the numerics were not present at all. The authors check that the convergence occurs with high probability when the hidden layer is sufficiently large---indeed, this is consistent with global convergence but I don't see how it probes anything directly related to the analysis. There is a lot of focus on the number of zeros in the trained preactivation (or inactive units). I do not immediately see how this is connected to the convergence results. I understand that the authors want to provide some contrast with Elkabetz and Cohen, but I am not sure that these experiments are really necessary for that. There is a large literature explaining the difference between the linear regime and nonlinear regime, including the papers already mentioned in the manuscript by Chizat.

---

> ### Author Response · Authors · 2023-02-04
> **Response to Reviewer Uxfn**
>
> We are very grateful your enthusiastic feedback, thank you!
> We have applied all the requested changes, please find below our comment on some of the mentioned issues.
>
> **Primer on DI.**
> We have tried to improve parts of the presentation in Section 2 to make it more approachable.
>
> **Numerical experiments -- initial motivation.**
> We decided on not analyzing the scaling regimes from Theorem 5.6 in our work as the experiments presented already suggest that a better bound of order $\mathcal{O}(N/d_0)$ should hold empirically.
> We therefore decided on presenting the results that would allow for better comparison with other related works (Oymak--Soltanolkotabi, 2020 and Elkabetz--Cohen, 2021).
> Regarding the number of numerical zeros -- apart from allowing us to compare more easily with Elkabetz--Cohen, 2021, we believe that this finding is of independent interest as it shows some new fundamental behavior of SGD iterates and thus we decided on paying it more attention in the main text.
>
>
> **Numerical experiments -- changes.**
> We have also tried to improve that part of the presentation, in particular by adding more explanation in the intro to Section 6.

---

### Decision · Action_Editors · 2023-02-27

**Recommendation:** Accept as is

**Comment:**

Reviewers agreed that this is a technically interesting work, which contributes to existing literature by providing an important improvement that further closes the gap between upper and lower bound on this basic problem. While some parts of the proof technique appeared in earlier works (e.g. Nguyen et al, Bianchi et al), the authors appropriately acknowledge the existing literature. It is therefore recommended for acceptance at TMLR.


**Audience:**

Yes, this is intended for a general audience interested in the analysis of gradient descent algorithms on basic Neural Network architectures, starting from shallow ReLU networks.

**Claims And Evidence:**

This paper studies the minimal amount of overparametrisation needed in a shallow ReLU network to fit a finite dataset drawn from a subgaussian distribution. By leveraging a different proof technique based on differential inclusions in continuous time, the authors improve upon existing upper bounds to establish that a width of  $O(N^{1.25})$ is sufficient to fit $N$ training examples.